# Hidden Tenants: Microbiota of the Rhizosphere and Phyllosphere of *Cordia dodecandra* Trees in Mayan Forests and Homegardens

**DOI:** 10.3390/plants11223098

**Published:** 2022-11-15

**Authors:** Carla G. May-Mutul, Miguel A. López-Garrido, Aileen O’Connor-Sánchez, Yuri J. Peña-Ramírez, Natalia Y. Labrín-Sotomayor, Héctor Estrada-Medina, Miriam M. Ferrer

**Affiliations:** 1Departamento de Manejo y Conservación de Recursos Naturales Tropicales, Universidad Autónoma de Yucatán, Mérida 97313, Mexico; 2Unidad de Biotecnología, Centro de Investigación Científica de Yucatán, Mérida 97205, Mexico; 3Departamento de Ciencias de la Sustentabilidad, El Colegio de la Frontera Sur Unidad Campeche, Lerma 24500, Mexico

**Keywords:** agroforestry systems, 16S rRNA gene sequencing, domestication of fruit trees, microbial ecology, microbial communities, fungal ITS sequencing, MiSeq amplicon sequencing

## Abstract

During domestication, the selection of cultivated plants often reduces microbiota diversity compared with their wild ancestors. Microbiota in compartments such as the phyllosphere or rhizosphere can promote fruit tree health, growth, and development. *Cordia dodecandra* is a deciduous tree used by Maya people for its fruit and wood, growing, to date, in remnant forest fragments and homegardens (traditional agroforestry systems) in Yucatán. In this work, we evaluated the microbiota’s alpha and beta diversity per compartment (phyllosphere and rhizosphere) and per population (forest and homegarden) in the Northeast and Southwest Yucatán regions. Eight composite DNA samples (per compartment/population/region combination) were amplified for 16S-RNA (bacteria) and ITS1-2 (fungi) and sequenced by Illumina MiSeq. Bioinformatic analyses were performed with QIIME and phyloseq. For bacteria and fungi, from 107,947 and 128,786 assembled sequences, 618 and 1092 operating taxonomic units (OTUs) were assigned, respectively. The alpha diversity of bacteria and fungi was highly variable among samples and was similar among compartments and populations. A significant species turnover among populations and regions was observed in the rhizosphere. The core microbiota from the phyllosphere was similar among populations and regions. Forests and homegarden populations are reservoirs of the *C. dodecandra* phyllosphere core microbiome and significant rhizosphere biodiversity.

## 1. Introduction

Since the early colonization of terrestrial ecosystems, plants have maintained a close relationship with diverse microbial communities, mainly bacteria and fungi, which comprise the microbiota. These communities can be found in different plant compartments, such as roots (rhizosphere), stems (caulosphere), leaves (phyllosphere), flowers (anthosphere), fruits (carposphere), or seeds (spermosphere) [1,2]. Microbiota maintain a series of interactions with plant cells, benefiting growth, producing secondary metabolites, and protecting against pathogens in the host plant [2,3].

The composition and structure of the plant microbiota can present variations related, for example, to the genotype of the host and the geographic region where it develops [4,5]. During domestication, genotype selection and agroecosystem management generate changes in the composition of the crop-associated microbiota, often with a reduction in diversity [1,5,6]. This phenomenon has been widely studied in annual species with agronomic importance. However, tropical fruit trees represent a scarcely studied field except for the *Citrus* genus [7].

The rhizosphere is considered a complex habitat where the interactions between the edaphic microbiota and the host plant allow the development of a hyperdiverse microbiome [1,8]. Plants can change and recruit beneficial microbial communities at the boundary between soil and roots through root-type-specific metabolic properties, and positively shape their rhizosphere microorganisms [1,9,10]. Beneficial rhizosphere microbes protect plants against pathogens mainly through niche antagonism, resource competition, or microbial diversity [8]. Some rhizosphere microorganisms can colonize other plant compartments and, through selection, can establish in a new habitat, forming endophytes. Phyllosphere endophytes harbor a diverse community of microorganisms from other compartments, such as the phylloplane (epiphytic to the leaves), the rhizosphere, and the caulosphere [4,11]. The phyllosphere colonization is modulated by microenvironmental variations on the outer and inner layers of the epidermis, or the interstitial spaces considered the selective filters for establishing endophytic microbiota. The pattern of microbiota distribution is not uniform across all regions of a leaf surface. An increased density of trichomes and stomata is known to favor phyllosphere colonization and a higher abundance of some endophytic species [5,12,13]. 

Study of the microbiota structure has accelerated in the last decade through metagenomic studies using next-generation sequencing techniques and functional DNA analysis [14,15]. Metagenomic studies of plant species in which wild populations and agroforestry systems coexist in the same geographic areas have demonstrated their usefulness for understanding domestication processes [1,16]. In addition, studying the microbiota of wild and cultivated plants provides essential information for developing breeding programs for species of economic and agricultural interest [7,12,17]. The sympatric distribution of wild and cultivated populations is a common phenomenon in Mesoamerica, where herbaceous annuals, perennials, shrubs, and native trees are traditionally managed in agroforestry systems [18,19]. Mayan homegardens, also known as “solares”, are traditional agroforestry systems with different multipurpose trees, shrubs, and herbaceous plants maintained by a family unit [20]. In the Mayan homegarden, trees receive direct or indirect irrigation and occasional pruning, are more widely spaced than in forests, and the soils are cleared and fallowed [21,22]. Mayan homegardens are one of the principal agroforestry systems in which Yucatec Maya communities have domesticated several edible species, including fruit trees [23,24].

*Cordia dodecandra* A. DC. (Cordiaceae) is a native tree species from medium-sized forests of Mesoamerica and is a structural species of the Mayan homegarden, so it is frequently found in these agroforestry systems. However, forest populations in the area have decreased because of deforestation and fragmentation [22]. Fruits are prepared in a conserve and more often consumed during Hanal-Pixan (Mayan festivities). The species is also valuable for its wood [23,24]. Previous studies were conducted to determine the domestication process of *C. dodecandra* in the Northeast and Southwest regions of the Yucatán Peninsula. Results indicate that the trees have larger leaves and flowers in homegardens, characteristics which are associated with particular genotypic groups, and that there is a high genetic flow between both regions [25]. Homegarden populations have similar genetic diversity to forest populations [26]. The populations from the Southwest region have a higher density of cytolytic pubescent trichomes than those from the Northeast regions [27]. Trees growing in homegarden populations have higher carbon and phosphorus concentrations in their leaves. Populations from the Northeast region have higher sodium and calcium content. Plants are established in sandier soils in the Northeast region. In homegarden populations, plants grow in soils with higher phosphorus content [28]. Collectively, these results suggest the presence of a domestication syndrome associated with the traditional management of this species in homegardens and with differences between the regions’ soils and between the forest and homegarden populations.

In this work we present the characterization of the microbiota of the rhizosphere and phyllosphere of *C. dodecandra* in wild and homegarden populations from the Northeast and Southwest regions of Yucatán. The amplicon MiSeq sequencing of 16S and ITS1-2 regions and the bioinformatic analysis from the sequences were used to evaluate whether the microbiota of bacteria and fungi has reduced structure and diversity in homegarden populations, because of the differential management that trees receive in these agroforestry systems, as well as to test whether microbial communities in the rhizosphere are more diverse than those in the phyllosphere. 

## 2. Results

### 2.1. Sequence Characteristics and Alpha Diversity

In respect of bacteria, 281,587 representative sequences (6810 to 55,520 per sample) were obtained, which, after exclusion of chloroplasts and mitochondria, totaled 107,946 (1043 to 45,314 per sample). In respect of fungi, 128,786 representative sequences (3959 to 25,893 per sample) were obtained. Due to variability in the number of representative sequences among samples, bacteria samples were normalized to 1000 and fungi to 3900 sequences, which are sufficient in order to infer differences in the microbiota diversity given that the rarefaction curves flatten out at a sequencing depth of 1000 for observed operating taxonomic units (OTUs) (Figure 1) and after sequencing depths of 500 and 1000 for the Shannon diversity index (entropy or information gain in the community) of bacteria and fungi, respectively (Appendix A Appendix A). The variation in alpha diversity estimates was considerable for bacteria: observed OTUs median 155, range 31 to 486; Chao1 index (estimated number of species in the community) median 160, range 31 to 845; and Shannon index median 4.61, range 3.09 to 5.83 (Figure 1). In respect of fungi, it was: observed OTUs median 180.5, range 51 to 212; Chao1 index median 213, range 51 to 239; and Shannon index median 2.90, range 1.86 to 3.54 (Figure 1). Therefore, no significant differences were observed between the bacterial and fungal microbiota of the phyllosphere and rhizosphere, or between forest and homegarden populations.

### 2.2. Taxonomic Assignation

The *C. dodecandra* microbiota accounts for a total of 618 and 1096 OTUs at the species level in the bacterial and fungal assignations, respectively. Taxonomic assignment was high (more than 80%) for taxa levels above family, less than 70% at genus and species level for bacteria, and less than 75% for all levels in fungi (Appendix A Appendix A). 

The majoritarian classes (those with >2.5% relative abundance per sample) were the Actinobacteria, Rubrobacteria, and Thermoleophilia, of the phylum Actinobacteriota; Aphaproteobacteria and Gammaproteobacteria of the phylum Proteobacteriota; Vicinamibacteria and Blastocatellia of the phylum Acidobacteriota; and Bacilli of the phylum Firmicutes (Figure 2). The minority classes (with <2.5% relative abundance per sample) comprised 60 from 27 different phyla (Appendix A Appendix A). The predominant phylum in the phyllosphere was Proteobacteriota, while in the rhizosphere Proteobacteriota and Firmicutes were the more predominant phyla. The variation in the relative abundance was notable among the samples from the different compartments and from the different populations from the order to the familial level (Appendix A Appendix A). 

At the species levels, in the phyllosphere, the majoritarian OTUs of bacteria were 9 out of 93 with a cumulative abundance of >70% per sample (Figure 3 and Appendix A Appendix A). The opposite case was observed in the rhizosphere, where 7 out of 560 were the majoritarian OTUs of bacteria with a cumulative abundance of <40% per sample (Figure 3 and Appendix A Appendix A). The majoritarian OTUs in the phyllosphere were Actinomycetospora uncultured bacteria, *Aureimonas jathrophae*, *Methylobacterium hispanicum*, *Methylobacterium komagatae*, and five unclassified OTUs from *Aureimonas*, *Allorhizobium-Neorhizobium-Pararhizobium-Rhizobium*, *Sphingomonas*, Sphingomonaceae, Rhizobiaceae, and Rhizobiales (Figure 3). In the rhizosphere, the majoritarian OTUs were Bacillus arbutinivorans, Microlunatus uncultured actinobacterium, and five unclassified OTUs of *Bacillus*, *Bradyrhizobium*, *Dongia*, Vicinamibcteriaceae, and Xanthobacteriaceae in the rhizosphere (Figure 3).

The majoritarian classes for fungi microbiota were the Eurotiomycetes, Dothideomycetes, and Sordariomycetes of the phylum Ascomycota, and Agaromycetes of the phylum Basidiomycota (Figure 4). The minority classes were 10 from the phyla Ascomycota, Basidiomycota, Chytridiomycota, and Mortierellomycota (Appendix A Appendix A). Most unclassified fungi came from the phyllosphere samples, particularly in the homegarden populations (Figure 4). The dominant classes were Ascomycota in the phyllosphere and Basidiomycota in the rhizosphere (Figure 4). From order level to genus, the relative abundance patterns were (i) similar in the phyllosphere samples from homegarden populations, which were represented by the same OTUs, and (ii) highly variable among the rhizosphere samples (Appendix A Appendix A).

In the phyllosphere, the majoritarian fungi OTUs were 5 out of the 485, with a cumulative abundance of more than 30% in three out of the four samples (Figure 5 and Appendix A Appendix A). In the rhizosphere, the majoritarian fungi OTUs were 9 out of the 784, with a cumulative abundance of more than 60% per sample (Figure 5 and Appendix A Appendix A). The majoritarian OTUs in the phyllosphere *were Strelitziana malaysiana*, an unclassified *Strelitziana*, and three unidentified OTUs of Basidiomycota, Ascomycota, and fungi (Figure 5). In the rhizosphere, the majoritarian fungi were *Nigrospora oryzae* and 11 unclassified OTUs of *Fomitopsis*, *Peniophora*, *Trechispora*, *Lepiota*, *Aspergillus*, Polyporales, Agaricales, Agaricomycetes, Basidiomycota, Ascomycota, and fungi (Figure 5).

### 2.3. Beta Diversity of the Microbiota

Differences in the Bray–Curtis distance values were significant between the phyllosphere and rhizosphere for bacterial microbiota (F_1,6_ = 3.79, P = 0.019, P_adj_ = 0.024) and fungal microbiota (F_1,6_ = 1.99, P = 0.025, P_adj_ = 0.036). However, no differences in those values were found for forest and homegarden populations in the bacterial microbiota (F_1,6_ = 0.63, P = 0.82, P_adj_ = 0.84) or fungal microbiota (F_1,6_ = 1.22, P = 0.28, P_adj_ = 0.31). The bacterial microbiota was grouped separately in quadrants (Cartesian notation): quadrant I, for the rhizosphere of the forest and the SW homegarden populations; quadrant III, for the phyllosphere of all populations; and quadrant IV, for the rhizosphere of the forest and the NE homegarden populations (Figure 6).

Similarly, in the heatmap, the bacterial microbiota was associated with two separate clusters integrating the phyllosphere and the rhizosphere samples (Figure 7). Within these clusters, the most similar samples were those from the forest and homegarden populations in the NE region (Figure 7). The OTU clusters for the phyllosphere microbiota were associated with two clusters. The first one was associated with enrichment of *Methylobacterium komagatae*, *Aureimonas jatrophae*, and an unclassified *Methylobacterium*—Phylum Proteobacteria, together with a non-culturable OTU, and an unclassified *Actinomycetospora*—phylum Actinobacteriota. The second cluster had an intermediate enrichment of unclassified OTUs of Streptomyces, 67-14, and one unculturable OTU—phylum Actinobacteriota, together with two unclassified OTUs and one unculturable OTU—phylum Proteobacteria, and an OTU of Vicinamibacteriaceae—phylum Acidobacteriota (Figure 7). The rhizosphere microbiota was associated with three clusters. The first one was shared with the second cluster of the phyllosphere microbiota, with enrichment of the same OTUs for the SW forest (Figure 7). The second cluster was enriched with unclassified OTUs of RB41—phylum Acidobacteria, *Dongia*—phylum Proteobacteria, together with Rubrobacter and a non-culturable *Mycrolunatus*—phylum Actinobacteriota—in the SW homegarden population (Figure 7). The third cluster was enriched with unclassified OTUs of Bacillus—phylum Firmicutes—and *Reynarella*, *Acidobacter*, *Stereidobacter*, and *Bradyrhizobium* of the phylum Actinobacteria in forests and homegarden populations in the NE (Figure 7).

For the fungal microbiota, the compartments and population samples were intermingled in the different quadrants: I, phyllosphere of the homegarden populations; II, rhizosphere of homegarden populations from both regions and NE forest; III, rhizosphere of the SW forest population; and IV, phyllosphere of homegarden populations (Figure 7). Except for the homegarden populations’ phyllosphere, all samples had variable differential enrichment and depletion patterns in the heatmap. Three clusters with highly enriched OTUs were identified. The first cluster was enriched with five unclassified OTUs of Chaetomiacea, *Chaetomium*, *Gaestrum*, Botryosphaeria, and Penicillium in the rhizosphere of the NW forest sample. The second cluster was enriched with *Fusarium solani*, *Cladosporium adianticola*, and two unclassified OTUs, one of Ceratobatisidiceae and one of Fungi, in the rhizosphere of the SW homegarden population. The third cluster was enriched with unclassified OTUs, two of Fungi, one of Micosphaerellaceae, and one of *Stretetziana* (Figure 8), from the phyllosphere of the SW forest. A fourth cluster had intermediate enrichment for OTUs of *Coletotrichum gloesporioides*, *Calopadia foliicola*, *Strelitziana malaysiana*, and an unclassified *Cyphellophora*, which was associated with the homegarden populations’ phyllosphere (Figure 8). The other three samples (SW forest rhizosphere, NW homegarden rhizosphere, and NE forest phyllosphere) had most of these OTUs depleted (Figure 8).

## 3. Discussion

In this work, to our knowledge, we report for the first time the microbiota of *C. dodecandra*, a Mesoamerican fruit tree. A total of 618 bacterial and 1096 fungal OTUs were recognized. More than half of these sequences had no taxonomic assignment, reflecting the lack of knowledge on the microbiota of tropical trees (particularly fruit trees) [3], as well as methodological constraints (e.g., choice of amplified regions, the database, or the pipeline used) [21,29]. The present study contributes to the characterization of the microbiota of homegardens in the Neotropics, in which native fruit species were domesticated by Mayan Yucatecan communities [23]. Comparative studies of microbiota from agroforestry systems are very scarce; as exceptions, we found reports of *Theobroma cacao* growing in diversified homegardens in Africa [30] and of *Citrus* growing in organic orchards in Brazil [31]. In these studies, many unknown OTUs with no taxonomic assignment were found, suggesting that there is a lack of knowledge in respect of the tropical microbiota in these systems. 

In general, the microbiota of cultivated plants in agricultural systems shows a decrease in biodiversity compared to wild populations, associated with introducing a few host genotypes, clonal propagation, monoculture, and no-rotation practices [1,17,32]. This effect was not observed in *C. dodecandra* since the relative abundance of the prevalent taxa among samples grouped by population was highly variable for the alpha-diversity indices. A similar pattern was found among the studied populations of *T. cacao* in Africa [30]. Mayan homegardens replicate the surrounding forests’ stratification and they maintain high agrobiodiversity, through soil and plant management, for food, medicine and ornamentation from the cultivated plants [20]. Therefore, all management practices of the soil and plants may contribute to large variability in the alpha- and beta-diversity of the rhizosphere, while the phyllosphere microbiota may be similar to that of the forest because plants are genetically similar to those of the nearby wild populations [26]. Our results suggest that homegarden populations may also be reservoirs of the phyllosphere’s forest microbiota since they harbor similar microbiota and biodiverse microbiota endophytes.

As expected, phyllosphere and rhizosphere microbiota presented differences in relative abundance and differential abundance patterns, with only a few shared OTUs (36 and 219 in bacteria and fungi, respectively). It has been proposed that the rhizosphere results from the selective recruitment of the edaphic microbiota and that the secretion of metabolites by the roots facilitates chemical communication between the microbial and host plant communities, leading to the consolidation of a symbiotic relationship between them [8,10,33]. In contrast, the endophytic microbiota of the phyllosphere is subject to intense selection due to the host plant’s immune system, secretion of cellular metabolites, and the phylloplane’s volatile and harsh environment. Generally, the phyllosphere microbiota is less diverse than that of other plant compartments, such as roots and stems [13,34,35]. In this study, diverse communities were found in the rhizosphere and phyllosphere for both bacteria and fungi, as observed in other domesticated plants in Mesoamerica, including *Agave* [17], maize [10], and tomato [36]. 

The phyllosphere’s microbiota had a lower species turnover than the rhizosphere’s microbiota, which may be associated with the presence of a core microbiome for *C. dodecandra* in the first compartment. In the phyllosphere, endophytes constitute a core microbiome that promotes atmospheric nutrient capture, foliar health, and conversion of growth by-products in wild populations [35,37] and in global-scale production of *Citrus* fruit crops [7]. However, in the rhizosphere, the high species turnover among all samples suggests that factors acting on a regional geographic scale (e.g., climate and soil origin) and on a local population scale (e.g., the identity of plant species from the neighborhood, management practices, and their intensity) may enhance the variability of the microbiota. The soil physicochemical properties and nutrient availability differ among the studied regions [28], as do the associated plants and management practices for the species in forest and homegarden populations from each region [38,39]. The rhizosphere’s microbiota from forest and homegarden populations were more similar within each region, suggesting that geographic variation has an essential effect on the distribution and abundance of the different taxa, as was observed in *Agave* [17]. The species turnover in rhizosphere microbiota has been associated with variability in soil characteristics and variation in the species assemblage of neighborhood plants, as well as genotype and morphological differences among hosts in various tropical trees [3,33]. Although composite samples analyzed in this study preclude assessing the individual-tree variation, it is also feasible that developmental or genetic characteristics of the host may shape the assemblages of *C. dodecandra* rhizosphere microbiota. To understand factors contributing to rhizosphere microbiota variation, further studies are required to analyze a more robust spectrum of samples from different populations and different edaphic and environmental conditions.

As for annual and perennial horticultural species, fruit tree species maintain interactions with the rhizosphere and phyllosphere microbiome that impact their growth, development, and health [1,2,9,40]. The most abundant bacterial taxa in the phyllosphere of *C. dodecandra* were the genera *Methylobacterium*, *Aureimonas,* and *Actinomycetospora*. *Methylobacterium* fixes atmospheric nitrogen and uses methanol (CH₃OH) or methane (CH₄) of plant origin, facultatively, as a source of carbon and energy. The bacteria that inhabit the phyllosphere favor colonization and can promote the growth and development of plants [10,41,42]. The recently described *Actinomycetospora* and *Aureimonas* genera are involved in carbon and nitrogen cycling [43,44]. Other majoritarian genera were *Allorhizobium–Neorhizobium–Parhizobium–Parhizobium–Rhizobium*, and *Sphingomonas*, which are also considered diazotrophic and key organisms in plant growth [45]. The bacterial OTU with the highest relative abundance in the rhizosphere was *Bacillus arbutinivorans*, which can solubilize phosphate and produce indole acetic acid in vitro, and when it is in a consortium with other *Bacillus* and *Streptomyces* it can increase the drought tolerance in poplar [46]. Some species from the genus *Microlunatus* and the Propionibacteriales order have dissimilatory nitrate reduction [47] and may accumulate polyphosphates [48]. Most of the identified bacteria taxa in the phyllosphere and rhizosphere of *C. dodecandra* are related to beneficial bacterial species that confer a higher fitness to the host plant.

The phyla Ascomycota and Basidiomycota were the more abundant for the fungal microbiota of *C. dodecandra*. These phyla have been previously reported as components of the fungal communities in the phyllosphere from tropical forest and agroforestry systems, with the phylum Ascomycota being dominant [30,34]. This pattern was observed in *C. dodecandra* phyllosphere microbiota. The majoritarian OTUs in the phyllosphere were the genera *Strelitziana* and *Neostrelitziana*, which cause leaf spots. The presence of these species in visually healthy trees suggests that they are latent saprotrophs, spreading once the tissue is dead, or that other species of bacteria or fungi associated with *C. dodecandra* may be antagonists to those pathogens. Further analysis of the functional networks of the microbiome of *C. dodecandra* may confirm that some bacteria may inhibit the growth of pathogens, as has been suggested [49,50,51]. In the rhizosphere, the predominance of Basidiomycota was not expected, because, in general terms, Ascomycota is the predominant phylum in the Yucatán soils [52], and in the rhizospheres of *Citrus* [7] and *Agave* [17], two tropical broadly-cultivated species. However, in the rhizosphere of beech, OTUs of the phylum Basidiomycota are the most abundant [53]. It has been proposed that Basidiomycota can contribute to lignin degradation when they are enriched in the rhizosphere of maize [54]. Therefore, the prevalence of Basidiomycota may have a similar function in the forest and agroforestry systems that *C. dodecandra* inhabits. The most abundant genera found in this study were lignin and cellulose degraders, *Fomitopsis*, *Trechispora* [55], *Peniophora* [56], and *Lepiota* [57]. Together with the other Agaricomycetes and Basidiomycota species, they contribute to transforming the polyaromatic compounds in the *C. dodecandra* rhizosphere. Several species of the genus *Aspergillus* have synergistic effects with mycorrhizae, which help promote plant growth [46], even in soils contaminated with heavy metals [58]. A great diversity of fungi associated with the rhizosphere, among the most abundant taxa, could be explained by the fact that they facilitate plant nutrition by transforming soil organic matter.

*Cordia dodecandra* is a native fruit tree that contributes to the food sovereignty of the Maya people when it is traditionally managed with other multipurpose species in the Mayan homegarden [20,22]. The caducifolious nature of the species may position this fruit tree as a key source of the microbiota and nutrients in the soil that are available to the plant. Homegarden populations may be considered the reservoirs of the bacterial and fungal species that form the core microbiome of *C. dodecandra*. However, further work is needed to understand whether the different microbiota communities recorded in this study are taxonomically diverse but functionally similar, and whether micro- and macro-environmental factors contribute to the large variability that this species has maintained during its domestication process. 

## 4. Materials and Methods

### 4.1. Study Area

The forest and homegarden *C. dodecandra* populations analyzed in the Northeast region are in the Tizimin municipality, and those in the Southwest region in the municipality of Tzucacab, both in Yucatán, Mexico (Figure 9). In each type of population (wild and homegarden), 12 adult individuals were selected with a diameter at breast height greater than 25 cm and an approximate height of between seven and ten meters, all with mature foliage and healthy appearance.

In the Northeast region, the climate is characterized as warm sub-humid with summer rainfall, of lower humidity (69.07%), and very warm and warm semi-dry (30.93%), with a mean annual temperature range of 24 to 26 °C and average precipitation of 600 to 1500 mm (Aw₁, according to Köeppen classification modified for Mexico by García) [59]. The predominant vegetation type is grassland (47%) and medium sub-deciduous forest (47.16%) [38]. In the homegarden, the predominant tree species that accompany *Cordia dodecandra* are mainly fruit species, such as *Citrus aurantium* and *Spondias purpurea* [22]. 

In the Southwest region, the climate is warm sub-humid with summer rains, of lower humidity (97.54%), and warm sub-humid with summer rains, of average humidity (2.46%). The average temperature oscillates between 24 and 28 °C, and the average precipitation is 1000 to 1200 mm (Awo’, according to Köeppen classification modified for Mexico by García) [59]. The predominant vegetation type (78.36%) is medium sub-deciduous and medium seasonal evergreen forest [38]. The tree species accompanying *Cordia dodecandra* in the homegarden are *Brosimum alicastrum*, *Manilkara zapota*, *Swietenia macrophylla*, and *Cedrela odorata* [39].

### 4.2. Sample Collection and Storage

Mature leaves were randomly collected from the canopies of trees. Samples in forest populations were obtained from 10 individuals in the Northeast region and 11 in the Southwest region. In the homegardens, samples from eight individuals in each region were obtained. The collected leaves were superficially washed with 70% ethanol, stored in sterile plastic bags, and transported on ice in a cooler to the facilities of El Colegio de la Frontera Sur Campeche unit’s facility, where they were stored at −80 °C for the subsequent extraction of metagenomic DNA.

The rhizosphere soil was obtained from the fine roots of three individuals from each population type and region. These samples were not taken from the edges of extensive paths in the forest populations or heavily trafficked areas in the homegarden. Rhizospheres were extracted by vigorously shaking the fine roots until they had no more loose soil, and then obtaining the attached rhizosphere soil by gentle brushing. They were transported on ice in a cooler to the facilities of the Centro de Investigación Científica de Yucatán unit’s facility, where they were stored at −80 °C for the subsequent extraction of metagenomic DNA.

### 4.3. DNA Extraction

The leaf surface was cleaned with 70% ethanol, and the tissue was macerated in liquid nitrogen. DNA extraction was performed with the ZymoBIOMICS™ DNA Miniprep kit (ZYMO RESEARCH; Irvine, CA, USA) following the protocol proposed by the manufacturer. Leaf DNA was concentrated by precipitation with 10 M ammonium acetate and resuspended in DNA-free pure water. DNA concentration was quantified using a Thermo Scientific Multiskan GO model FI-01620 spectrometer (Thermo Fisher Scientific, Waltham, MA, USA) with μDrop plate and SkanIt version 4.1 software (Thermo Fisher, Scientific, Finland). Based on the concentration of each sample, aliquots were taken to make a composite mixture by population type and region (giving a total of four for leaves). The rhizosphere soil of the three sampled trees was combined into a composite sample for each population type per region. DNA was extracted with the ZymoBIOMICS™ DNA Miniprep kit (ZYMO RESEARCH; Irvine, CA, USA) following the manufacturer’s protocol. Following elution, DNA samples were concentrated by ethanol precipitation and resuspended in 100 mL of free DNAse water.

The phyllosphere and rhizosphere composite samples were sent to RTL Genomics (Research and Testing Laboratories, Lubbock, TX, USA) for sequencing on the MiSeq Illumina platform (Illumina, San Diego, CA, USA) following the manufacturer’s protocol. For the bacterial microbiome, the universal primers 27F (5′-AGAGAGTTTGATCCTGGCTCAG-3′) and 338R (5′-GCTGCCTCCCGTAGGAGT-3′) were used for the 16S rRNA regions [60,61] and for the fungal microbiome the ITS1-2 regions with primers ITS1F (5′-CTTGGTCATTTAGAGGAAGTAA-3′) and ITS2aR (5′-GCTGCGTTCTTCATCGATGC-3′) were used [62,63].

### 4.4. Bioinformatics Analysis

The sequence processing was carried out using QIIME2 version 2022.2 [64] to obtain the taxonomic classification of the microbiome. The pipelines are presented in Appendix A and consisted of commands to evaluate (i) the quality of the forward and reverse sequences for the 16S rRNA gene and ITS1-2 regions, with the fastqc and multiqc algorithms [65]; and (ii) the sequence demultiplexing and quality control, with DADA2 [66]. The representative sequences of bacteria and fungi were obtained separately, and the clean sequences per sample were organized in a feature table. The taxonomic assignment was carried out using Silva’s 138-99-nb database for bacteria and Unite for fungi [64]. For the phyllosphere samples, filtering was performed to exclude 16S gene sequences corresponding to the host’s chloroplasts and mitochondria. The feature table and taxonomy matrices for bacteria (filtered) and fungi were imported into R, where a phyloseq diversity analysis was performed [67] using a pipeline (Appendix A) to conduct the following analyses. First, alpha diversity was compared between compartments and populations for the diversity indices (i) observed OTUs, (ii) estimated richness Chao1 [68], and Shannon diversity index [69], after rarifying the data [70]. The rarefaction curves flatten out after a sequencing depth of 1000 for the number of observed species of bacteria and fungi (Figure 1) and after sequencing depths of 500 and 1000 for the Shannon index of bacteria and fungi, respectively (Appendix A Appendix A), proving the sufficiency of the data. Subsequently, differences in the relative abundances of taxa—from phylum to species—by compartment (rhizosphere and phyllosphere) and by population (forest and homegarden) were characterized for the non-normalized data, and plots were obtained for the majority taxa (those with relative abundance per sample >2.5% of OTUs). Beta diversity was characterized using principal component analysis using Bray–Curtis distances [71] and comparing these with an Adonis test to obtain F values and the associated P and adjusted *p* value (P_adj_) with a PermANOVA. A bias-corrected microbiome composition analysis ANCOM-BC [72] using the data from the differential abundance analysis of OTUs [73] and the heat maps was obtained for the 20 most abundant OTUs to graphically analyze the similarity between OTU compartments, populations, and phyla [74].

## 5. Conclusions

The characterization of the microbiota associated with wild ancestors and traditional agroforestry systems represents a valuable source for further studies on functional microbiome diversity to understand domestication and attain sustainable agricultural systems [16,45,75]. The conservation of microbiota associated with native fruit trees with biocultural importance, such as *C. dodecandra*, is a key element for preserving the resilience of the forest and homegarden populations where they grow. Indirect effects of the management of the homegardens and the actual distribution of the species may contribute to the high species turnover in the rhizosphere microbiota, while genetic similarity and the beneficial roles of endophytes may explain the similarity among the phyllosphere microbiota. Future research should be focused on the key factors contributing to the high diversity of *C. dodecandra* microbiota and its bioprospection.

## Figures and Tables

**Figure 1 plants-11-03098-f001:**
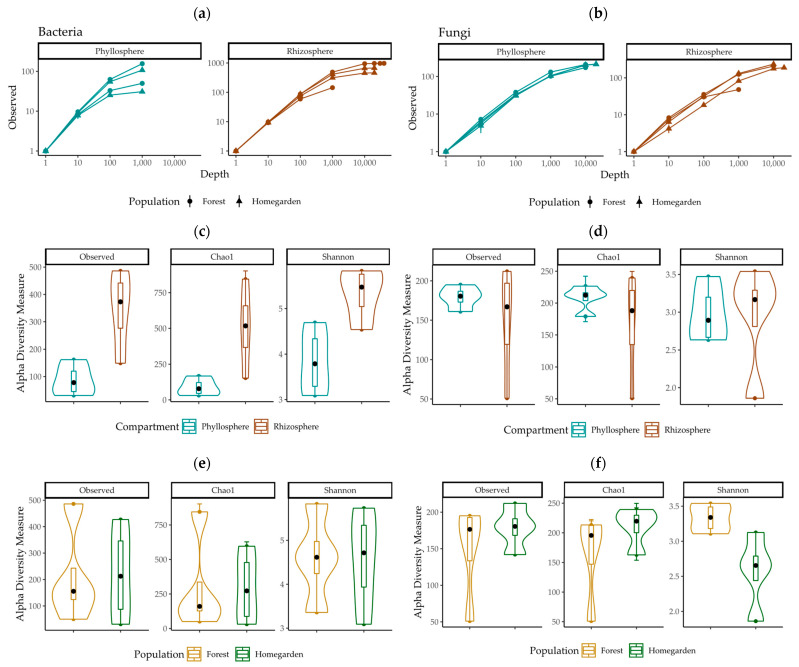
Rarefaction curves depicting the observed frequency of operating units (OTUs) by the sequencing depth in (**a**) bacteria, (**b**) fungi, and the alpha-diversity measures in (**c**,**e**) bacteria and (**d**,**f**) fungi microbiota from (**c**,**d**) phyllosphere and rhizosphere, in (**e**,**f**) Yucatán’s forest and homegarden populations of *Cordia dodecandra*.

**Figure 2 plants-11-03098-f002:**
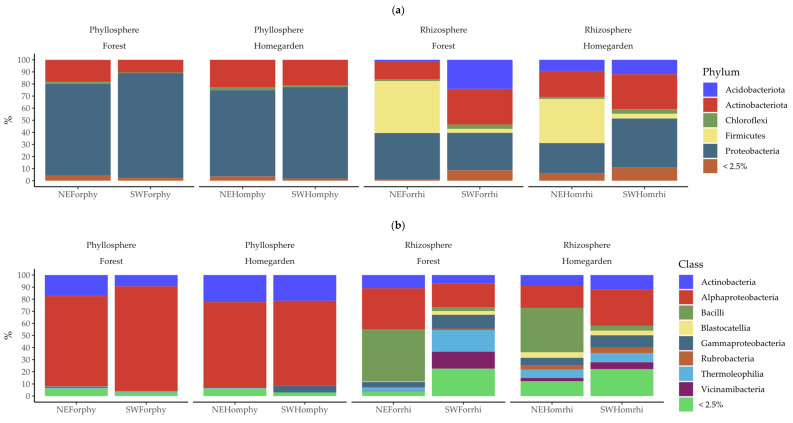
Relative frequency of (**a**) phylum and (**b**) class assigned to 618 bacteria OTUs from phyllosphere and rhizosphere microbiota in Yucatán’s forest and homegarden populations of *Cordia dodecandra*.

**Figure 3 plants-11-03098-f003:**
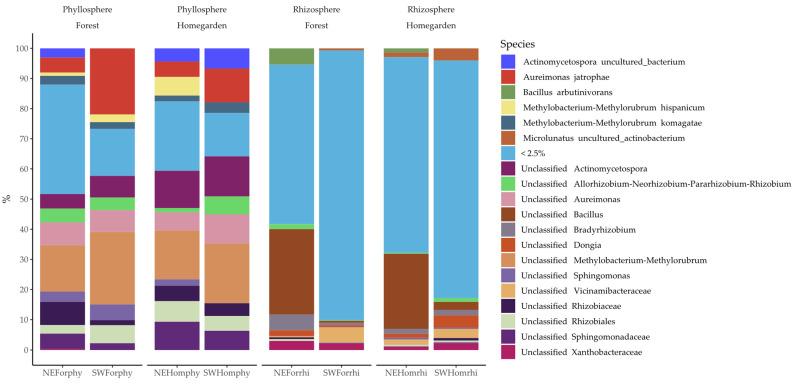
Relative frequency of 618 bacteria OTUs from phyllosphere and rhizosphere microbiota in Yucatán’s forest and homegarden populations of *Cordia dodecandra*.

**Figure 4 plants-11-03098-f004:**
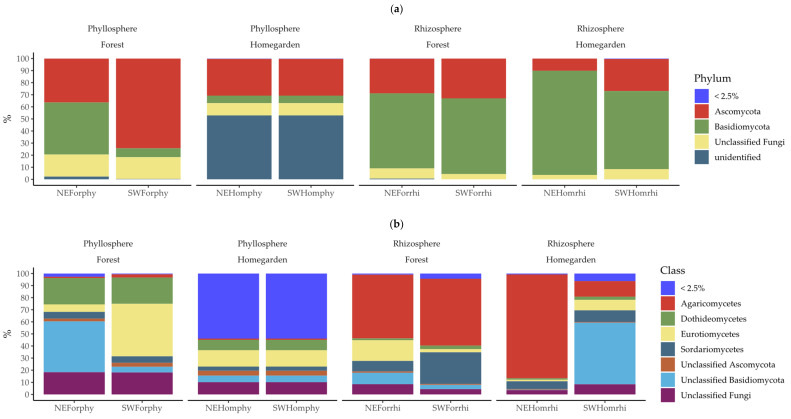
Relative frequency of (**a**) phylum and (**b**) class assigned to 1096 fungi OTUs from phyllosphere and rhizosphere microbiota in Yucatán’s forest and homegarden populations of *Cordia dodecandra*.

**Figure 5 plants-11-03098-f005:**
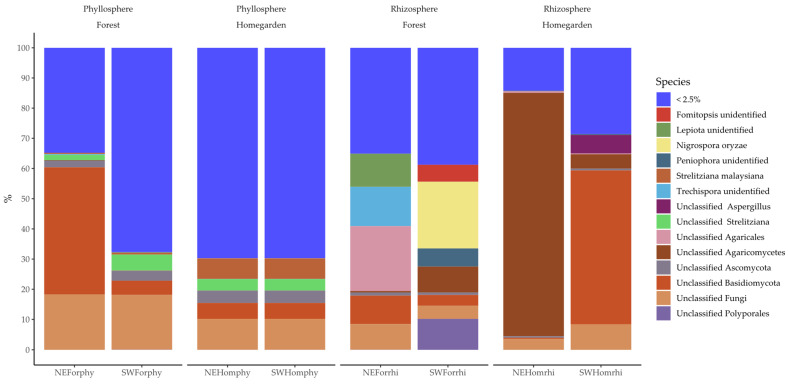
Relative frequency of 1096 fungi OTUs from phyllosphere and rhizosphere microbiota in Yucatán’s forest and homegarden populations of *Cordia dodecandra*.

**Figure 6 plants-11-03098-f006:**
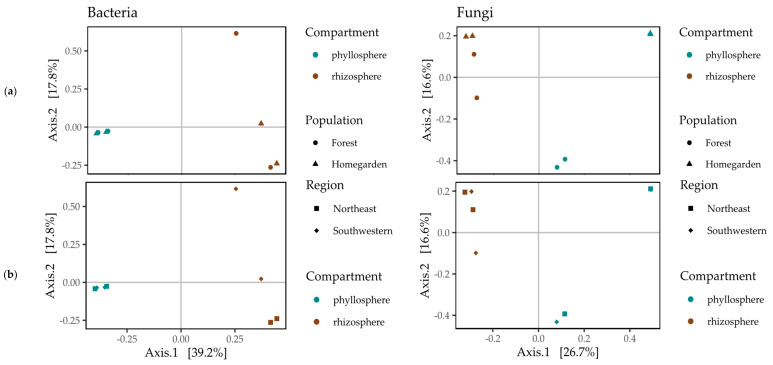
Principal component analysis of the microbiota communities from the phyllosphere and rhizosphere in (**a**) forest and homegarden populations of *Cordia dodecandra* in (**b**) the Northeast and Southwest regions of Yucatán.

**Figure 7 plants-11-03098-f007:**
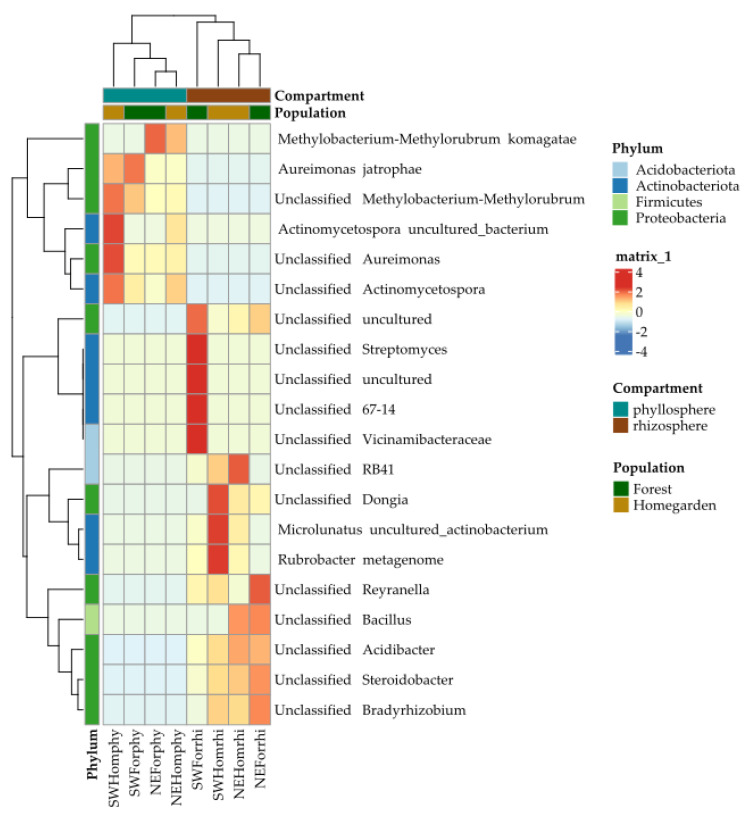
Heatmap of the microbiota of bacteria from the phyllosphere and rhizosphere in Yucatán’s forest and homegarden populations of *Cordia dodecandra*.

**Figure 8 plants-11-03098-f008:**
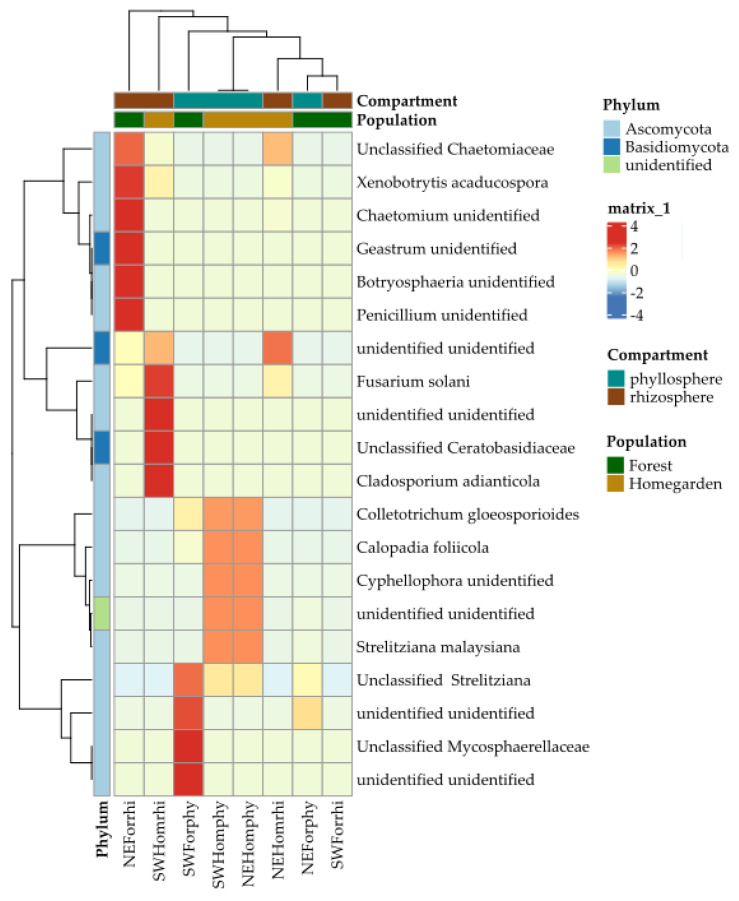
Heatmap of the fungal microbiota from the phyllosphere and rhizosphere in Yucatán’s forest and homegarden populations of *Cordia dodecandra*.

**Figure 9 plants-11-03098-f009:**
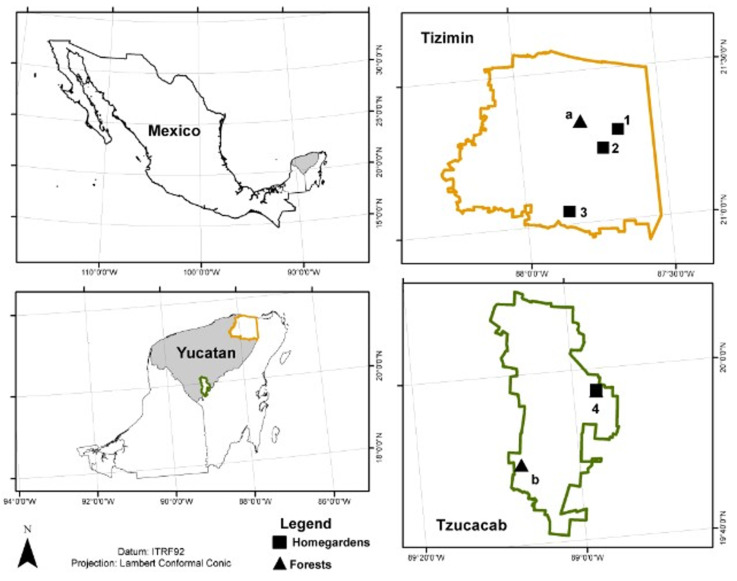
Geographic location of *Cordia dodecandra* populations from (**a**, **b**) forests and (**1**–**4**) homegardensin the (**a**, **1**–**3**) Northeast region (Tizimín municipality) and (**b**, **4**) Southwest region (Tzucacab municipality).

## Data Availability

The sequences from which these data were obtained can be found in NCBI-Genbank (Appendix B).

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
