# Peer review of "Hidden Tenants: Microbiota of the Rhizosphere and Phyllosphere of Cordia dodecandra Trees in Mayan Forests and Homegardens"

_plants, 2022, doi:10.3390/plants11223098_

Round 1
Reviewer 1 Report
Title: Hidden tenants: microbiota of the rhizosphere and phyllosphere of Cordia dodecandra trees in Mayan forests and homegardens
Authors: C. G. May-Mutul, M. Á. López-Garrido, A. O'Connor-Sánchez, Y. J. Peña-Ramírez, N. Y. Labrín-Sotomayor, H. Estrada-Medina, M. Monserrat Ferrer
Reference: plants-1975968
Article type: Research
Reviewer Comments:
The manuscript plants-1975968, entitled “Hidden tenants: microbiota of the rhizosphere and phyllosphere of Cordia dodecandra trees in Mayan forests and home gardens”, studies the composition and compares the microbiome from the rhizosphere and phyllosphere of C.t dodecandra trees in Mayan forests and home gardens.
Despite the lack of innovative features, the topic covered in the present papers is important and targets a broad range of readers presenting implications in several fields of knowledge.
General comments:
In the reviewer's opinion, the authors of this paper do not have English as their native tongue. A high degree of editing and revising will be necessary before the manuscript can be appropriately reviewed, as there are numerous statements whose meaning is unclear, and the text is convoluted and hard to read.
Due to the extension of the corrections needed, instead of presenting the suggestion by line, the reviewer presents the bulk sections of the text (suggestions in bold).
The Abstract is comprehensive and well-structured. Although, the sub-heading should be removed.
The keywords should be presented in alphabetic order.
The Introduction section is quite long and includes non-essential data that should be removed for the simplification of the text.
The Result section is deeply confusing and could not be revised. The majority of the sentences lack sense. The text was only revised for grammar but it needs to be re-written.
The Discussion section is deeply confusing and could not be revised. The majority of the sentences lack sense. The text was only revised for grammar but it needs to be re-written.
The Material and Methods needs to be simplified. If the ordinary protocols were used, there is no need to describe them. It is enough to cite the reference and point out if modifications were made.
The Conclusion section covers general concepts. An effort should be made to present conclusion related to the presented work.
Specific comments:
Lines 22-37: please consider replacing it with
During domestication, the selection of cultivated plants often reduces microbiota diversity. Microbiota in compartments such as the phyllosphere or rhizosphere can promote health, growth, and development from fruit trees. Cordia dodecandra, a deciduous tree used by Mayan civilizations for its fruits and wood, grows in remnant forest fragments and home gardens from Yucatan.
This work evaluates the microbiota’s alpha- and beta-diversity per compartment (phyllosphere-endophytes and rhizosphere) and population (forest and home gardens) in the Northeast and Southwest Yucatan regions. Eight composite DNA samples (per compartment/population/region combination) were amplified for 16S-RNA (Bacteria) ITS1-2 (Fungi) and sequenced by Illumina MiSeq. Bioinformatic analyses were performed with Qiime and phyloseq.
For Bacteria and Fungi, from 107,947 and 128,786 assembled sequences, 618 and 1,092 OTUs were assigned, respectively. Alpha-diversity of Bacteria and Fungi was highly variable among samples and similar among compartments and populations. A significant species turnover among populations and regions was observed in the rhizosphere. The core microbiota from the phyllosphere was similar among populations and regions.
Forests and home gardens are reservoirs of the C. dodecandra phyllosphere core microbiome and significant rhizosphere biodiversity.
Lines 41-116: please consider replacing it with
Since the early colonization of terrestrial ecosystems, plants have maintained a close relationship with diverse microbial communities, mainly bacteria and fungi, which comprise the microbiota. These communities can be found in different plant compartments, such as roots (rhizosphere), stems (caulosphere), leaves (phyllosphere), flowers (anthosphere), fruits (carposphere), or seeds (spermosphere) [1, 2]. Microbiota maintains a series of interactions with plant cells, benefiting growth, producing secondary metabolites, and protecting against pathogens in the host plant [1-3]. The composition and structure of the plant microbiota can present variations related, for example, to the host's genotype and the geographic region where it develops [4, 5]. During domestication, genotype selection and agroecosystem management generate changes in the composition of the crop-associated microbiota, often with a reduction in diversity [2, 5, 6]. This phenomenon has been widely studied in annual species with agronomic importance. However, tropical fruit trees represent a scarcely studied field, except for the Citrus genus [7].
The rhizosphere is a complex habitat where the interactions between the edaphic microbiota and the host plant allow the development of a hyperdiverse microbiome [2, 8]. Plants can change and recruit beneficial microbial communities at the boundary between soil and roots through root-type-specific metabolic properties and positively shape rhizosphere microorganisms [2, 9, 10]. Beneficial rhizosphere microbes protect plants against pathogens mainly through niche antagonism, resource competition, or microbial diversity [8]. Some rhizosphere microorganisms can colonize other plant compartments and, through selection, can establish in a new habitat, forming endophytes. Phyllosphere endophytes harbor a diverse community of microorganisms from other compartments, such as the phylloplane (epiphytic to the leaves), rhizosphere, and caulosphere [4, 11]. The phyllosphere colonization is modulated by microenvironmental variations on the outer and inner layers of the epidermis, or the interstitial spaces considered the selective filters for establishing endophytic microbiota. The pattern of microbiota distribution is not uniform across all regions of a leaf surface. An increased density of trichomes and stomata is known to favor phyllosphere colonization and a higher abundance of endophytic species [5, 12, 13].
The study of microbiota structure accelerated in the last decade through metagenomic studies using next-generation sequencing techniques and functional DNA analysis [14, 15]. Metagenomic studies of plant species in which wild populations and agroforestry systems coexist in the same geographic areas have demonstrated their usefulness for understanding domestication processes [2, 16]. In addition, studying the microbiota of wild and cultivated plants provides essential information for developing breeding programs for species of economic and agricultural interest [7, 12, 17]. The sympatric distribution of wild and cultivated populations is a common phenomenon in Mesoamerica, where herbaceous annuals, perennials, shrubs, and native trees are traditionally managed in agroforestry systems [18, 19]. Mayan home gardens, also known as "solares", are traditional agroforestry systems with different multipurpose trees, shrubs, and herbaceous plants maintained by a family unit [20]. In home gardens, trees receive direct or indirect irrigation and occasional pruning, are more widely spaced than in forests, and the soils are cleared and fallowed [21, 22]. Home gardens are one of the principal agroforestry systems in which the Yucatec Maya communities have domesticated several edible species, including fruit trees [23, 24]. Cordia dodecandra A. DC. (Cordiaceae) is a native tree species from medium-sized forests of Mesoamerica and is a structural species of the Mayan home garden, frequently found in these agroforestry systems. However, the forest populations in the area have decreased because of deforestation and fragmentation [22]. Fruits are prepared in a conserve and often consumed during Hanal-Pixan (Mayan festivities). The species is also valuable for its wood [23, 24].
Previous studies were conducted to understand the domestication process of C. dodecandra in the Northeast and Southwest regions of the Yucatan Peninsula. Results indicate that the trees have larger leaves and flowers in home gardens, characteristics associated with particular genotypic groups, and that there is a high genetic flow between both regions [25]. Home garden populations have similar genetic diversity to forest populations. The populations from the Southwest region have a higher density of cytolytic pubescent trichomes than those from the Northeast regions [26]. Home garden populations have higher carbon and phosphorus content on their leaves. Populations from the Northeast region have higher sodium and calcium content. Plants are established in sandier soils in the Northeast region. In home garden populations, plants grow in soils with higher phosphorus content [27]. Collectively, these results suggest the presence of a domestication syndrome associated with the traditional management of this species in home gardens and with differences between the regions' soils and for the forest and home garden populations.
This work presents the characterization of the microbiota of the rhizosphere and phyllosphere of C. dodecandra in wild and home garden populations from the Northeast and Southwest regions of Yucatan. The amplicon Miseq sequencing of 16S and ITS1-2 regions and the bioinformatic analysis from the sequences was used to evaluate if the microbiota of bacteria and fungi has a reduced structure and diversity in home gardens because of the differential management that trees receive in these agroforestry systems, as well as to test if microbial communities in the rhizosphere are more diverse than in the phyllosphere.
Lines 119-257: please consider replacing it with
2. Results
2.1. Sequence characteristics and alpha diversity
In Bacteria, 281,587 sequences (6,810 to 55,520 per sample) were obtained. After excluding chloroplastidial and mitochondrial DNA, 107,946 sequences (1,043 to 45,314 per sample) remained.
In Fungi, 128,786 sequences (3,959 to 25,893 per sample) were obtained. Due to the variability in the number of reads between samples, Bacteria samples were normalized to 1000 and Fungi to 3900 sequences (Figure 1). The variation in alpha-diversity estimates was considerable for Bacteria: observed OTUs (median 155, range 31 to 486), Chao1 (median 160, range 31 to 845), and Shannon indices (median 4.61, range 3.09 to 5.83) (Figure 2), as well as in Fungi: observed OTUs (median 180.5, range 51 to 212), Chao1 (median 213, range 51 to 239) and Shannon indices (median 2.90, range 1.86 to 3.54) (Figure 1). Therefore, no significant differences were observed between bacterial and fungal microbiota of phyllosphere and rhizosphere or between forest and home garden populations.
Figure 1. Rarefaction curves depicting the observed frequency of OTUs by sequencing depth and alpha-diversity measures from phyllosphere and rhizosphere microbiota in Yucatan's forest and home garden populations of Cordia dodecandra.
Taxonomic classification and relative abundance
A total of 618 and 1096 OTUs were grouped at the species level in the bacterial and fungal microbiota, respectively. Taxonomic assignment was high (more than 80%) for taxa levels above family, less than 70% at genus and species level for Bacteria, and less than 75% for all levels in Fungi (Supplementary material T1 and T2). The phyla with > 2.5% relative abundance for the bacterial microbiota of C. dodecandra were Proteobacteria with two classes, Actinobacteriota with six classes, Acidobacteriota with ten classes, Chloroflexi with ten classes, and Firmicutes with two classes. The minority phyla with < 2.5% relative abundance were 22 with 48 different classes (Figure 2, Supplementary material T1).
Figure 2. Relative frequency of phylum and class assigned to 618 bacteria OTUs from phyllosphere and rhizosphere microbiota in Yucatan's forest and home garden populations of C. dodecandra.
In the phyllosphere, 12 of the 93 Bacteria OTUs had a relative abundance of >2.5%, with a cumulative abundance of >70% per sample (Figure 3). The opposite was observed in the rhizosphere, where 547 of the 560 Bacteria OTUs had a relative abundance of <2.5%, with a cumulative abundance of >40% per sample (Figure 3).
Figure 3. Relative frequency of 618 bacteria OTUs from phyllosphere and rhizosphere microbiota in Yucatan's forest and home garden populations of C. dodecandra.
The genera (family-order) Actinomycetospora (Pseudonocardiaceae-Pseudonocardiales), Allorhizobium-Neorhizobium-Parhizobium-Rhizobium, Aureimonas (Rhizobiaceae-Rhizobiales), Methylobacterium-Methylorubrumy (Beijerinckiaceae-Rhizobiales), Sphingomonas (Sphingomonadaceae-Sphingomonales) are prevalent in the phyllosphere, and Bacillus (Bacillaceae-Bacillales), an unclassified genus (Xanthobacteriaceae-Rhizobialles), and minority taxa (<2.5% of relative abundance) are prevalent in the rhizosphere (Supplementary material F1-F3). Finally, the taxa having >2.5% of relative abundance are Actinomycetospora uncultured bacteria, Aureimonas jathrophae, Methylobacterium hispanicum, and Methylobacterium komagatae in the phyllosphere Bacillus arbutinivorans, Microlunatus uncultured acinobacterium, and two unculturable OTUs of Bacillus and Dongia in the rhizosphere; five unculturable OTUs, while the OTUs are having relative abundance <2.5 % add up to more than 10% and 40% of cumulative abundance per sample in the phyllosphere and rhizosphere respectively (Figure 3). The taxa with a relative abundance of >2.5% for Fungi microbiota were the phyla Ascomycota, with eight classes, and Basidiomycota, with three classes; while the minority taxa were Chytridiomycota and Mortierellomycota, with one class each (Figure 4, Supplementary material T2).
Figure 4. Relative frequency of phylum and class assigned to 1096 fungi OTUs from phyllosphere and rhizosphere microbiota in Yucatan's forest and home garden populations of C. dodecandra.
In the phyllosphere, 5 out of the 485 Fungi OTUs had a relative abundance of >2.5%, with a cumulative abundance of more than 30% in three of the four samples (Figure 5). In the rhizosphere, 9 out of the 784 Fungi OTUs had a relative abundance of >2.5%, with a cumulative abundance of more than 60% per sample (Figure 6).
Figure 5. Relative frequency of 1096 fungi OTUs from phyllosphere and rhizosphere microbiota in Yucatan's forest and home garden populations of C. dodecandra.
Most unclassified fungi came from the phyllosphere samples, particularly in the home garden populations (Figure 5). The prevalent classes, with a relative abundance of >2.5%, were Eurotiomycetes, Dothideomycetes, Sordariomycetes from the phylum Ascomycota, Agaricomycetes from the phylum Basidiomycota, three unclassified taxa and all taxa a relative abundance of <2.5% (Figure 4). The phyllosphere and rhizosphere were dominated by Ascomycota and Basidiomycota (Figure 4). From order level to genus, the relative abundance patterns were: i) similar in the samples of phyllosphere from home gardens, which the same OTUs; and ii represented) highly variable among the rhizosphere samples (Supplementary material F1-F3). Taxa with a relative abundance < 2.5% each were the most abundant in the phyllosphere, along with Strelitziana malaysiana and unidentified Strelitziana (fam. Inserta cedis-Chaetothyriales) and three unidentified OTUs of Basidiomycota, Ascomycota, and Fungi (Figure 6). In the rhizosphere, the taxa with a relative abundance of >2.5% were Nigrospora oryzae (Trichosphaeriaceae-Trichosphaeriales), five unclassified OTUs of the genera Fomitopsis (Fomitopsidaceae-Polyporales), Peniophora (Peniophoraceae-Russulales), Trechispora (Hydnodonthaceae-Trechisporales), Lepiota (Agaricaceae-Agaricales), Aspergillus (Aspergillaceae-Eurotiales), two belonging to the orders Polyporales and Agaricales, and four unidentified OTUs from Agaricomycetes, Basidiomycota, Ascomycota, and Fungi (Figure 4).
2.2. Beta diversity of the microbiota
Differences in Bray-Curtis distance values were significant between phyllosphere and rhizosphere for bacterial microbiota (F1,6 = 3.79, P = 0.019, Padj = 0.024) and fungal microbiota (F1,6 = 1.99, P = 0.025, Padj = 0.036). However, no differences in those values were found for forest and home garden populations in bacterial microbiota (F1,6 = 0.63, P = 0.82, 207 Padj = 0.84) or fungal microbiota (F1,6 = 1.22, P = 0.28, Padj = 0.31). The Bacterial microbiota was grouped separately in quadrants (Cartesian notation): I, for the rhizosphere of the forest and the SW home garden populations; III, for the phyllosphere of all populations; and IV, for the rhizosphere of the forest and the NE home garden populations (Figure 6).
Figure 6. Principal Component Analysis of the microbiota communities from phyllosphere and rhizosphere in the forest and home garden populations of C. dodecandra in the Northeast and Southwest regions of Yucatan.
Similarly, in the heatmap, the bacterial microbiota was associated in two separate clusters integrating the phyllosphere and the rhizosphere samples each (Figure 7). Within these clusters, the most similar samples were those from the forest and home gardens in the NE region (Figure 7). The OTU clusters for the phyllosphere microbiota were associated with two clusters. The first one, with enrichment of Methylobacterium komagatae, Aureimonas jatrophae, an unclassified Methylobacterium -phylum Proteobacteria; together with a non-culturable OUT, and an unclassified Actinomycetospora - phylum Actinobacteriota. The second cluster has an intermediate enrichment of unclassified OTUs of Streptomyces, 67-14, and one unculturable - phylum Actinobacteriota, together with two unclassified and one unculturable OTUs - phylum Proteobacteria, and an OTU of Vicinamibacteriaceae - phylum Acidobacteriota (Figure 7). The rhizosphere microbiota was associated with three clusters. The first cluster shared with the second cluster of the phyllosphere microbiota, with enrichment of the same OTUs for the SW forest (Figure 7). The second cluster enriched with unclassified OTUs of RB41 - phylum Acidobacteria, Dongia - phylum Proteobacteria - together with Rubrobacter and a non-culturable Mycrolunatus - phylum Actinobacteriota- in the SW home gardens (Figure 7). The third cluster enriched with unclassified OTUs of Bacillus - phylum Firmicutes - and Reynarella, Acidobacter, Stereidobacter, and Bradyrhizobium - phylum Actinobacteria - in forests and home gardens in the NE (Figure 7).
Figure 7. Heatmap of the microbiota of Bacteria from phyllosphere and rhizosphere in the forest and home garden populations of Yucatán's C. dodecandra.
For the fungal microbiota, the compartments and populations samples were intermingled in the different quadrants: I) phyllosphere of the home garden populations; II) rhizosphere of home garden populations from both regions and NE forest; III) rhizosphere of the SW forest population; IV) phyllosphere of home garden populations (Figure 7). Except for the home gardens' phyllosphere, all samples had differential enrichment and depletion patterns in the heatmap. Three clusters with highly enriched OTUs were identified. The first cluster with five unclassified OTUs of Chaetomiacea, Chaetomium, Gaestrum, Botryosphaeria, and Penicillium in the rhizosphere of the NW forest sample. The second cluster with two unclassified OTUs of Fungi, one of Ceratobatisidiceae, Fusarium solani, and Cladosporium adianticola, in the rhizosphere of the SW home gardens. The third cluster with unclassified OTUs, two Fungi, one of Micosphaerellaceae and one of Stretetziana (Figure 8) from the phyllosphere of the SW forest. A fourth cluster had intermediate enrichment for OTUs of Colletotrichum gloesporioides, Calopadia foliicola, Strelitziana malaysiana, and an unclassified Cyphellophora associated with the home gardens' phyllosphere (Figure 8). The other three samples (SW forest rhizosphere, NW home garden rhizosphere, and NE forest phyllosphere) had most of these OTUs depleted (Figure 8).
Figure 8. Heatmap of the fungal microbiota from phyllosphere and rhizosphere in the Yucatán's forest and home garden populations of C. dodecandra.
Lines 259-378: please consider replacing it with
3. Discussion
In this work, to our knowledge, we report for the first time the microbiota of C. dodecandra, a Mesoamerican fruit tree. A total of 618 bacterial and 1096 fungal OTUs were obtained. More than half of these sequences had no taxonomic assignment, reflecting the lack of knowledge on the microbiota of tropical trees (particularly fruit trees) and methodological constraints (e.g., choice of amplified regions, the database, or the pipeline used) [28, 29]. Studies in agroforestry systems are very scarce. As an exception, the reports of Theobroma cacao growing in diversified home gardens in Africa [30] and Citrus growing in organic orchards in Brazil [31]. In these studies, many unknown OTUs with no taxonomic assignment were found. Therefore, the present study contributes to the characterization of the microbiota of agroforestry systems, such as home gardens in the Neotropics, with native fruit species domesticated by Mayan Yucatecan communities [23].
In general, the microbiota of cultivated plants shows a decrease in biodiversity associated with introducing a few host genotypes, monoculture and no-rotation management, and clonal propagation in agricultural systems [2, 17]. This effect was not observed in C. dodecandra since the relative abundance of the prevalent taxa among samples was highly variable, with diversity indices - observed richness, Chao1, and Shannon - grouped by population were also highly variable. Home gardens' diversity replicates the surrounding forests' stratification, where the agrobiodiversity is maintained by the families that use many of these species for food, medicine, or ornamentation [20]. Incorporating beneficial native and exotic species may promote increased species diversity of the rhizosphere microbiota. However, as discussed below, it may have a minor impact on the phyllosphere, where a core microbiota is maintained for the species. Results suggest that home gardens, under traditional management conditions, are reservoirs of the forest microbiota since they harbor microorganisms as biodiverse as those of forest populations. As expected, phyllosphere and rhizosphere microbiotas presented differences in relative abundance and differential abundance patterns, with only a few shared OTUs (36 and 219 in Bacteria and Fungi, respectively) due to different selection pressures for microorganisms. In general, a few taxa can establish themselves as endophytes. It has been proposed that the rhizosphere results from the selective recruitment of the edaphic microbiota and that the secretion of metabolites by the roots facilitates chemical communication between microbial and the host plant communities, leading to the consolidation of a symbiotic relationship between them [8, 10, 32].
In contrast, the phyllosphere endophytic microbiota is subjected to intense selection due to the host plant's immune system, secretion of cellular metabolites, and the phylloplane's volatile and harsh environment. Generally, the phyllosphere microbiota is less diverse than other plant compartments, such as roots and stems. [13, 33, 34]. In this study, diverse communities were found in the rhizosphere and phyllosphere for both Bacteria and Fungi, as observed in other domesticated plants in Mesoamerica, including Agave [17], maize [10], and tomato [35].
Species turnover between communities for the phyllosphere microbiota was low due to a core microbiome for C. dodecandra. The presence of a core microbiome for phyllosphere endophytes has been proposed. This microbiome is associated with atmospheric nutrient capture, foliar health, and conversion of growth by-products in wild populations [34, 36].
On the other hand, species turnover for rhizosphere microbiota among forest and home garden populations from the two studied regions suggest factors leading to variabilities (e.g., climate and soil origin) on a regional geographic scale. Additionally, those like the identity of plant species from the neighborhood, management practices, and their intensity may act on a population scale. The soil physicochemical properties and nutrient availability differ among the studied regions [27], as do the associated plants and management practices the species had in forest and home garden populations from each region [37-39]. The association of rhizospheres of forest and home garden populations by region observed for the fungal microbiota suggests that geographic variation has an essential effect on the distribution and abundance of the different microbiota taxa, as was observed in Agave [17].
The species turnover in rhizosphere microbiota has been associated with variability in soil characteristics, variation in the species assemblage of neighborhood plants, genotype, and morphological differences among hosts in various tropical trees [3, 32]. Although composite samples analyzed in this study preclude assessing the individual-tree variation, it is also feasible that developmental or genetic characteristics of the host may also shape the assemblages of C. dodecandra rhizosphere microbiota. To understand the factors contributing to rhizosphere microbiota variation, further studies are required to analyze a more robust spectrum of samples from different populations and edaphic and environmental conditions.
As for annual and perennial horticultural species, fruit tree species maintain interactions with the rhizosphere and phyllosphere microbiome that impact their growth, development, and health [1, 2, 9, 40].
The most abundant bacteria in the C. dodecandra phyllosphere were Methylobacterium and Aureimonas (order Rhizobiales, class Alphaproteobacteriam phylum Proteobacteria) and Actinomycetospora (order Pseudonocardiales, class Actinobacteria, phylum Actinobacteriota). Other prevalent taxa with a relative abundance >2.5% were Allorhizobium, Neorhizobium, Parhizobium, Rhizobium (order Rhizobiales, family Rhizobiaceae), and Sphingomonas (order Sphingomonadales, family Sphingomonadaceae).
Methylobacterium fixes atmospheric nitrogen and uses methanol (CH₃OH) or methane (CH₄) of plant origin as a source of carbon and energy. These bacteria inhabit the phyllosphere, favor colonization, and can promote plants' growth and development [10, 41, 42].
The recently described Actinomycetospora and Aureimonas genera are involved in carbon and nitrogen cycling [43, 44].
Most of the identified bacteria in C. dodecandra phyllosphere and rhizosphere are related to beneficial species that confer a higher fitness to the host plant.
The bacterial OTU with the highest relative abundance in the rhizosphere was Bacillus arbutinivorans. These bacteria solubilize phosphate and produce indole acetic acid in vitro and, when it is in consortium with other Bacillus and Streptomyces species, increase the drought tolerance in poplar [45].
Some species from the genus Microlunatus and the order Propionibacteriales had dissimilatory nitrate reduction [46] and may accumulate polyphosphates [47].
The phyla Ascomycota and Basidiomycota were prevalent in the fungal microbiota. These phyla have been previously reported as components of the fungal communities in the phyllosphere from tropical forest and agroforestry systems, with the phylum Ascomycota being the dominant [30, 33]. The prevalence of pathotrophic Fungi in the phyllosphere and the rhizosphere of C. dodecandra is suggested by the high relative abundance of Chaetothyriales. The genera Strelitziana and Neostrelitziana cause leaf spots, and Nigrospora oryzae causes rot. These species in visually healthy trees suggest that they are latent saprotrophs, spreading once the tissue is dead or that other species of bacteria or fungi associated with C. dodecandra may be antagonists to those pathogens.
Further analysis of the functional networks of the C. dodecandra microbiome helps to understand if some bacteria may inhibit the growth of pathogens, as has been suggested [48-50]. In the rhizosphere, the predominance of Basidiomycota was not expected. Generally, Ascomycota is the predominant phylum in the Yucatan soils [51], in the rhizospheres of Citrus [7], and Agave [17], two broadly-cultivated tropical species. However, in the rhizosphere of beech, OTUs of the phylum Basidiomycota are the most abundant [52]. It has been proposed that these fungi can contribute to lignin degradation when enriched in the rhizosphere of maize [53]. Therefore, the prevalence of Basidiomycota may have a similar function in the forest and agroforestry systems in which C. dodecandra inhabits. The most abundant genera found in this study are the lignin and cellulose degraders Fomitopsis, Trechispora [54], Peniophora [55], and Lepiota [56]. Together with other Agaricomycetes and Basidiomycota species, they contribute to transforming the polyaromatic compounds in the C. dodecandra rhizosphere. Several species of the genus Aspergillus have synergistic effects with mycorrhizae, which help promote plant growth [45], even in soils contaminated with heavy metals [57]. A great diversity of fungi associated with the rhizosphere, among the most abundant taxa, could be explained by the fact that they facilitate plant nutrition by transforming soil organic matter.
Lines 379-468: please consider replacing it with
4. Materials and Methods
4.1. Study area
The forest and home garden C. dodecandra populations analyzed in the Northeast region are located in the Tizimin municipality, and those in the Southwest region in Tzucacab municipality (Yucatan, Mexico) (Figure 9). In each type of population (wild and home garden), 12 adult individuals were selected with a diameter at breast height greater than 25 cm and an approximate height of between seven and ten meters, all with mature foliage and a healthy appearance.
In the Northeast region, the climate is characterized as warm sub-humid with summer rainfall Aw₁ of lower humidity (69.07%) and very warm and warm semi-dry (30.93%); with a mean annual temperature range of 24 to 26°C and average precipitation of 600 to 1 500 mm. [58]. The predominant vegetation type is grassland (47%) and medium sub-deciduous forest (47.16%) [37]. In the home garden, the predominant tree species accompanying C. dodecandra are mainly fruit species, such as Citrus aurantium and Spondias purpurea [38].
In the Southwest region, the climate is warm sub-humid with summer rains Awo' of lower humidity (97.54%) and warm sub-humid with summer rains, of average humidity (2.46%), the average temperature oscillates between 24 and 28°C, and the average precipitation is 1 000 to 1 200 mm [58]. The predominant vegetation type (78.36%) is medium sub-deciduous and medium seasonal evergreen forest [37]. The tree species accompanying C. dodecandra in the home gardens are Brosimum alicastrum, Manilkara zapota, Swietenia macrophylla, and Cedrela odorata [39].
4.2. Sample collection and storage
Mature leaves were randomly collected from the canopy of trees. Samples in forest populations were obtained from ten individuals in the Northeast region and 11 in the Southwest region. In home gardens, eight individuals in each region were obtained.
Collected leaves were superficially washed with 70% ethanol, stored in sterile plastic bags, and transported on ice to the El Colegio de la Frontera Sur Campeche unit's facilities, where they were stored at -80°C for subsequent extraction of metagenomic DNA.
The rhizosphere soil was obtained from fine roots of three individuals from each population type and region. These samples were not found at the edges of extensive paths in the forest populations or heavily trafficked areas in the home gardens. Rhizospheres were extracted by vigorously shaking the fine roots until they had no more loose soil and then obtaining the attached rhizosphere soil by gentle brushing. They were transported on ice to the Centro de Investigación Científica de Yucatán unit's facility, where they were stored at -80°C for subsequent extraction of metagenomic DNA.
4.3. DNA extraction
The leaf surface was cleaned with 70% ethanol, and the tissue was macerated in liquid nitrogen. DNA extraction was performed with the kit ZymoBIOMICS™ DNA Miniprep (ZYMO RESEARCH; Irvine, CA, USA) following the protocol proposed by the manufacturer. Leaf DNA was concentrated by precipitation with 10M ammonium acetate and resuspended in DNA-free pure water. DNA concentration was quantified using a Thermo Scientific Multiskan GO model FI-01620 spectrometer with μDrop plate and SkanIt version 4.1 software. Based on the concentration of each sample, aliquots were taken to make a composite mixture by population type and region (giving a total of 4 for leaves). The rhizosphere soil of the three sampled trees was combined into a composite sample for each population type per region. DNA was extracted with the ZymoBIOMICS™ DNA Miniprep (ZYMO RESEARCH; Irvine, CA, USA) following the manufacturer protocol.
Following elution, DNA samples were concentrated by ethanol precipitation and resuspended in 100 l pure water.
Phyllosphere and rhizosphere samples were sent to RTL Genomics for sequencing on the MiSeq Illumina platform, following the manufacturer's protocol.
For the bacterial microbiome, the universal primers 27F (5'-AGAGAGTTTGATCCTGGCTCAG-3') and 338R (5'-GCTGCCTCCCGTAGGAGT-3') were used for the 16S rRNA regions [59, 60] and for the fungal microbiome the ITS1-2 regions with primers ITS1F (5'-CTTGGTCATTTAGAGGAAGTAA-3') and ITS2aR (5'-GCTGCGTTCTTCATCGATGC-3') [61, 62].
4.4. Bioinformatics analysis
The sequence processing was carried out using QIIME2 version 2022.2 [63] to obtain the taxonomic classification of the microbiome. The pipelines are presented in Supplementary materials File1, which consisted of powers to evaluate:
i) the quality of the forward and reverse sequences for the 16S rRNA gene and ITS1-2 regions, with the fastqc 444 and multiqc algorithms [64]
ii) the sequence demultiplexing and quality control, with 445 DADA2 [65].
The representative sequences of Bacteria and Fungi were obtained separately, and the clean sequences per sample were organized in a feature table. The taxonomic assignment was done using Silva's 138-99-nb database for Bacteria and Unite for Fungi [63]. For the phyllosphere samples, filtering was performed to exclude 16S gene sequences corresponding to the host's chloroplasts and mitochondria. The feature table and taxonomy matrices for Bacteria (filtered) and Fungi were imported into R, where a phyloseq diversity analysis was performed [66] using a pipeline (Supplementary materials File2) to conduct the following analyses. First, alpha diversity was compared between compartments and populations after homogenizing for differences in sampling effort (number of reads) and rarifying the data [67]. This normalization was done because the objective is to compare alpha diversity patterns between compartments and populations for the diversity indices: i) observed OTUs, ii) estimated richness Chao1 [68], and Shannon diversity index [69]. Subsequently, differences in the relative abundance of taxa - from phylum to species - compartment (rhizosphere and phyllosphere) and population (forests and home gardens) were characterized for the non-normalized data, and plots were obtained for the majority taxa (those with relative abundance per sample >2.5% of OTUs). Beta diversity was characterized using principal component analysis using Bray Curtis distances [70] and comparing these with an Adonis test to obtain F values and the associated P and adjusted P value (Padj) with a PermANOVA. A bias-corrected microbiome composition analysis ANCOM-BC [71] with the data from the differential abundance analysis of OTUs [72] and the heat maps were obtained for the 20 most abundant OTUs to graphically analyze the similarity between OTU compartments, population, and phyla [73].
Lines 469-479: please consider replacing it with
5. Conclusions
The present work contributes to the critical study of C. dodecandra, a fruit tree traditionally managed in Maya-Yucatecan home gardens. These agroforestry systems are reservoirs for this native species. Together with forests, these systems harbor diverse bacterial and fungal species that form the core microbiome of C. dodecandra.
The predominant phyllosphere microbiota confers multiple benefits to its host, as documented for other native Mesoamerican and fruit species. The hyperdiverse associations with rhizosphere microbiota have a high species turnover among forest and home garden populations from different regions. These associations are explained as plastic responses to micro and macro environmental factors that need to be explored to understand the domestication process in tropical fruit trees.
Scientific comments:
Line 195 and 202: Please verify the use of “Fungi”. Please consider being more precise.
Author Response
The authors thank the reviewer for the careful revision of our manuscript. The comments that we received are very valuable and represent an opportunity to improve the communication of our results.
Please find below the list of changes or rebuttals for each point that was raised.
As a separate document, we send you the current manuscript containing the requested modifications and editing for the English language and style.
Specific comments:
The sub-heading of the abstract was removed (lines 22-37).
The keywords are presented in alphabetic order (lines 38-39).
We include all the grammar suggestions except for:
- The name “homegarden” was kept instead of “home garden” because the first is used to denote the traditional agroforestry system that is managed by a familiar unit, while the second is used to denote a piece of land adjacent to a house where plants are grown. After the edition of language and style, the editors approve this term as adequate.
- We refer to Maya people instead of Mayan civilizations, to highlight that the persons in the communities (past and present) are the ones managing Cordia dodecandra (lines 25 and 411-412).
General comments
The Introduction section was rewrittenen (lines 44-115). We decided to keep all information about the domestication process of the species (lines 88 -107) because the invitation to publish this paper was done for the special number “Advances in Domestication of Fruit Trees”.
The Result section, including the figure legends, was summarized and re-written (lines 116-298)
The Discussion section was organized and re-written (lines 300-420).
In the material and methods, all protocols were kept for DNA extraction and bioinformatic analyses to provide the methods and equipment needed for the repeatability of the metagenomic analyses conducted in this species (lines 464-516).
The Conclusion section was re-written and the focus was set on the results of this research (lines 518-528).
The use of Bacteria and Fungi terms as Kingdoms and common names was checked throughout the manuscript.
Reviewer 2 Report
Dear authors,
The study of plants microbiomes is a promising research field. In this paper, the microbiota’s alpha, and beta diversity per compartment (phyllosphere-endophytes and rhizosphere) and per population (forest and homegardens) from Northeast and Southwest Yucatan regions were evaluated. The study of plants microbiomes is a promising research field. Plant-microrganisms symibiosis play a key role in plant health and synthesis of secondary metabolites. I appreciate the hard work and the effort spent in the study. The overall research is interesting and worth publishing in Plants. However, before getting accepted, the following comments should be taken in consideration.
1) Line 50: “the host’s genotype of the host” Please, correct.
2) C. dodecandra in lines 137, 140, 473 should be italic.
3) Please carefully revise the reference
· Some references have P. refering to the pages number while some references without P.
· Refernces without a volume and pages e.g. refernce 22 52(2), 140–152.
· Refernces without a year e.g. refernce 56 the year is 2009.
Author Response
The authors thank the reviewer for the revision of our manuscript. The comments that we received are very valuable and represent an opportunity to improve the communication of our results.
Please find below the list of changes or rebuttals for each point that was raised.
As a separate document, we send you the current manuscript containing the requested modifications and editing for the English language and style.
1) Line 50: Correction was included “the genotype of the host” was kept.
2) lines 1 37, 140, 473 C. dodecandra font is in italics now.
3) Reference section and cites within the main text were edited and now all of them are in a similar and complete format.
Reviewer 3 Report
In the manuscript entitled "Hidden tenants: microbiota of the rhizosphere and phyllosphere of Cordia dodecandra trees in Mayan forests and homegardens" the authors describe the alpha and beta diversity of the Cordia dodecandra microbiota. Illumina MiSeq data was obtained from various populations as well as from different parts of the plant. 16S-RNA sequence analysis was used to assess the bacterial community, and ITS1-2 sequence analysis was used for fungi, which corresponds to the widely used microbiota analysis technique.
The study of the plant microbiome is a relevant and interesting area in modern science. This work is certainly interesting and worthy of publication, however, I have a few comments.
I am very confused by the total number of sequences 107,947 (Bacteria) and 128,786 (Fungi) for a sufficiently large number of samples analyzed in this work, this is clearly not enough for the analysis of the community of fungi and bacteria; the authors also did not prove the sufficiency of their data.
It is necessary to improve the informativeness of figures 2,3,4,5. Indicate in the figures the percentage characteristic for each element of the column. For an illustrative example, you can refer to the article (https://doi.org/10.3390/plants11091128) in which the design of the drawings is quite well developed.
Author Response
The authors thank the reviewer for the careful revision of our manuscript. The comments that we received are very valuable and represent an opportunity to improve it.
Please find below the list of changes or rebuttals for each point that was raised.
- The total number of sequences 107,947 (Bacteria) and 128,786 (Fungi) analyzed in this work for the phyllosphere are sufficient to conduct the analyses, differences in the microbiota diversity given that the rarefaction curves flatten out at a sequencing depth of 1000 for observed operating taxonomic units (OTUs) (Figure 1) and after sequencing depths of 500 and 1000 for the Shannon index of bacteria and fungi, respectively (Supplementary material F1). Thanks to the call of the reviewer, we summarize the information in the result section lines 123-127 and the materials and methods and include it as well in the materials and methods section lines 503- 506.
- To improve the informativeness of the figures depicting the relative abundance we change the continuous color palette to a discrete one, the rows in which facet data was presented, and the orientation of the page to visualize at once the different patterns among compartments and population samples. As suggested by the reviewer, and replicating the figures on the suggested paper, we include the percentage of each OTU on the cells, but due to the large diversity of the phyllosphere and rhizosphere communities of Cordia dodecandra, the information of the different labels was overlapped. We include now the relative abundance for each sample in the Supplementary materials Table 1 and Table 2, as a reference for the readers.
As a separate document, we send you the current manuscript containing the requested modifications and editing for the English language and style.
Reviewer 4 Report
Dear Authors,
Your research article deals with the interesting topic of microbiota of the rhizosphere and phyllosphere of Cordia dodecandra trees in Mayan forests and homegardens. The language appears to be correct, but I don't feel qualified to judge about the English language and style. It is prepared carefully. However, I have a few comments. Some are debatable:
1. Due to the layout of the manuscript, I believe that in the introduction the abbreviations "OTUs", "Chao1" and "Shannon Diversity Index" should be clarified. Information about them appears only at the end of the manuscript. It should be remembered that the reader is not always a specialist in this field.
2. Line 222. It should be OTU instead of OUT.
3. What is the practical application of your research? Please emphasize this in the summary.
Good luck!
Sincerely yours
Reviewer
Author Response
The authors thank the reviewer for the careful revision of our manuscript. The comments that we received are very valuable and represent an opportunity to improve the communication of our results.
Please find below the list of changes or rebuttals for each point that was raised.
- The abbreviation OTUs was defined the first time it was mentioned in the abstract (line 32) results section (line 125) and the figure legends (line 144). The explanation of the diversity indexes Chao1 and Shannon were also included the first time they were mentioned in the results section(lines 129 and 126 respectively)
- Line 259 (Line 222 before). Corrected OTU instead of OUT
- Practical applications of the research were included in the discussion section (lines 411-420) and the conclusions (lines 518-528).
As a separate document, we send you the current manuscript containing the requested modifications and editing for the English language and style.
Round 2
Reviewer 1 Report
The manuscript was significantly improved and is now ready for publication.